# Chaining Mutual Information and Tightening Generalization Bounds

**Amir R. Asadi**[1][*]    **Emmanuel Abbe**[1,2]    **Sergio Verdú**

[1]Princeton University    [2]EPFL

## Abstract

Bounding the generalization error of learning algorithms has a long history, which yet falls short in explaining various generalization successes including those of deep learning. Two important difficulties are (i) exploiting the dependencies between the hypotheses, (ii) exploiting the dependence between the algorithm's input and output. Progress on the first point was made with the chaining method, originating from the work of Kolmogorov, and used in the VC-dimension bound. More recently, progress on the second point was made with the mutual information method by Russo and Zou '15. Yet, these two methods are currently disjoint. In this paper, we introduce a technique to combine the chaining and mutual information methods, to obtain a generalization bound that is both algorithm-dependent and that exploits the dependencies between the hypotheses. We provide an example in which our bound significantly outperforms both the chaining and the mutual information bounds. As a corollary, we tighten Dudley's inequality when the learning algorithm chooses its output from a small subset of hypotheses with high probability.

## 1 Introduction

### 1.1 Motivation

Understanding the generalization phenomenon in machine learning has been a central question for many years and revived in recent years with the success and mystery of deep learning: why do neural nets generalize well, although they operate in a classically overparametrized setting? In particular, classical generalization bounds do not explain this phenomenon (see e.g. [1], [2]). Even simpler instances of successful machine learning problems and algorithms are not explained satisfactorily with current generalization bounds, e.g. [2]. This paper aims at deriving tighter generalization bounds for learning algorithms by combining ideas from information theory and from high dimensional probability.

Generalization bounds have evolved throughout the years, starting from the basic union bound over the hypothesis set, the refined union bound, Rademacher complexity, chaining and VC-dimension [3], [4]; and algorithm-dependent bounds such as PAC-Bayesian bounds [5], uniform stability [6], compression bounds [7], and recently, the mutual information bound [8].

We highlight two pitfalls among the key limitations of current bounds:

**A. Ignoring the dependencies between the hypotheses.** Consider the following example (which we refer to as Example I): an algorithm observes $G^2 = (G_1, G_2)$, where $G_1$ and $G_2$ are two independent standard normal random variables; the hypothesis set $\mathcal{H} = \{h_t : t \in T\}$ consists of functions $h_t(G^2) \triangleq \langle t, G^2 \rangle$, where $T \triangleq \{t \in \mathbb{R}^2 : \|t\|_2 = 1\}$. Suppose the algorithm is designed

---

[*]Corresponding author: `aasadi@princeton.edu`

to choose the hypothesis which achieves $\max_{t \in T} h_t(G^2)$. Since $h_t(G^2), t \in T$ are all zero mean random variables, the expected bias of the algorithm is $\mathbb{E}\left[\max_{t \in T} h_t(G^2)\right]$. Moreover, since $\mathcal{H}$ consists of an uncountable number of hypotheses, the union bound (or equivalently the maximal inequality) over the hypothesis set gives a vacuous bound. However, the fact is that we are not dealing with infinite number of *independent* random variables: the random variables $h_t(G^2)$ and $h_s(G^2)$ are actually quite dependent on each other when $t$ and $s$ are close.

To exploit the dependencies, the powerful technique of *chaining* has been developed in high dimensional probability in order to obtain uniform bounds on random processes, and has proven successful in a variety of problems including statistical learning. More specifically, chaining is the method for proving the tightest generalization bound using VC-dimension [9], [10]. Originating from the work of Kolmogorov in 1934 (see [9, p. 149]) and later developed by Dudley, Fernique, Talagrand and many others [11], the basic idea of chaining is to first describe the dependencies between the hypotheses by a metric $d$ on the set $T$, then to discretize $T$ and to approximate the maximal value ($\max_{t \in T} h_t(G^2)$) by approximating the maxima over successively refined finite discretizations, using union bounds at each step, and by introducing the notion of $\epsilon$-*nets* and *covering numbers* [12]. For instance, with this method, one can prove the finite upper bound $\mathbb{E}\left[\max_{t \in T} h_t(G^2)\right] \leq 19.0353$. Even for many examples of finite hypothesis sets, chaining is known to give far tighter bounds than the union bound [9]. Next we state a fundamental result which is based on the chaining method. For a metric space $(T, d)$, let $N(T, d, \epsilon)$ denote the covering number of $(T, d)$ at scale $\epsilon$. For the definitions of $\epsilon$-net and covering number, see Definition 8 in Section C of the supplementary material, and for the definition of seperable subgaussian processes see Definitions 1 and 2.

**Theorem 1** (Dudley). *[13]. Assume that $\{X_t\}_{t \in T}$ is a separable subgaussian process on the bounded metric space $(T, d)$. Then*

$$\mathbb{E}\left[\sup_{t \in T} X_t\right] \leq 6 \sum_{k \in \mathbb{Z}} 2^{-k} \sqrt{\log N(T, d, 2^{-k})}. \tag{1}$$

Note that PAC-Bayesian bounds, compression bounds and bounds based on uniform stability also do not exploit the dependencies between the hypotheses as they are not based on any metric on the hypothesis set.

**B. Ignoring the dependence between the algorithm input (data) and output.** Generalization bounds based on Rademacher complexity[2] and VC-dimension only depend on the hypothesis set and not on the algorithm, effectively rendering them too pessimistic for practical algorithms. Recent experimental findings in [1] have shown that in the over-parameterized regime of deep neural nets, such complexity measures give vacuous bounds for the generalization error. A possible explanation for that vacuousness is as follows: if $\mathcal{H} = \{h_t : t \in T\}$ denotes the hypothesis set and for every $t \in T$, $X_t$ denotes the generalization error of hypothesis $h_t$ and $W$ denotes the index of the chosen hypothesis by the algorithm, then to upper bound the expected generalization error $\mathbb{E}[X_W]$, one uses

$$\mathbb{E}[X_W] \leq \mathbb{E}\left[\sup_{t \in T} X_t\right], \tag{2}$$

and aims at upper bounding $\mathbb{E}\left[\sup_{t \in T} X_t\right]$ with these bounds, hence giving a *uniform bound* over the generalization errors of the *entire hypothesis set*. However, all we need to control is the generalization error of the *specific hypothesis $W$* selected by the algorithm. That expected generalization error of $W$ can be much smaller than the right side of (2) (see also [14]). In other words, such bounds are not taking into account the input-output relation of the algorithm, and uniform bounding seems to be too stringent for these applications. Consider the following example (which we refer to as Example II): let $X_1, X_2, ..., X_n$ be standard normal random variables and assume that the algorithm output is index $W$. Therefore the expected bias of the algorithm is $\mathbb{E}[X_W]$ and the goal is to upper bound it. By the maximal inequality (or equivalently the union bound), we have

$$\mathbb{E}\left[\sup_{1 \leq i \leq n} X_i\right] \leq \sqrt{2 \log n}, \tag{3}$$

where (3) is asymptotically tight if $X_i, i = 1, 2, ..., n$ are independent (see [12, Chapter 2]). But what if the algorithm is always more likely to choose $W$ among a small subset of $\{1, 2, ..., n\}$? Then $\mathbb{E}[X_W]$ could be much smaller than the right side of (3), as the chances of having an outlier value is smaller. Or, if the choice of $W$ is not dependent on the data, then $\mathbb{E}[X_W] = \mathbb{E}[\mathbb{E}[X_W|W]] = 0$. Interestingly, to explain this phenomenon and to obtain tighter upper bounds on $\mathbb{E}[X_W]$ an important information theoretic measure appears: the *mutual information*. This was originally proposed in the key paper of Russo and Zou [8] and then generalized in [16], [17], and in [18] for infinite number of hypotheses:

**Theorem 2.** *[8][18] Let $\{X_t\}_{t \in T}$ be a random process and $T$ an arbitrary set. Assume that $X_t$ is $\sigma^2$-subgaussian and $\mathbb{E}[X_t] = 0$ for every $t \in T$, and let $W$ be a random variable taking values on $T$. Then*

$$|\mathbb{E}[X_W]| \leq \sqrt{2\sigma^2 I(W; \{X_t\}_{t \in T})}. \tag{4}$$

In Example II, instead of using (2) and (3), one can have the tighter upper bound

$$\mathbb{E}[X_W] \leq \sqrt{2I(W; X_1, ..., X_n)}. \tag{5}$$

For example, if the algorithm chooses $W$ among $\{1, 2, ..., \lceil \log n \rceil\}$ with probability $1 - o(1)$, then (5) implies

$$\mathbb{E}[X_W] \leq \sqrt{2\left((1 - o(1)) \log(\log n) + o(1) \log(n - \log n) + 1\right)} \ll \sqrt{2 \log n}. \tag{6}$$

However, this method does not give a finite bound for Example I, since

$$I\left(\operatorname{argmax}_{t \in T} h_t(G^2); \{h_t(G^2)\}_{t \in T}\right) = \infty. \tag{7}$$

Similarly, as discussed in [19], the mutual information bound for perturbed SGD or any iterative algorithm which adds degenerate noise in each iteration blows up, and information-theoretic strategies for analyzing generalization error of such algorithms have not been reported.

## 1.2 This paper

By combining the ideas of the chaining method and the mutual information method, in this paper we obtain a *chained mutual information* bound on the expected generalization error which takes into account the dependencies between the hypotheses as well as the dependence between output and input of the algorithm. When applied to the two aforementioned simple examples (Examples I and II), our bound yields the better bound between the classical chaining and classical mutual information bounds. More importantly, we provide examples for which our bound outperforms both of the previous bounds significantly: in Example 1 we provide a family of cases where the chaining method gives a relatively large constant, the mutual information bound blows up, but our bound tends towards zero. We also discuss how our new bound gives a possible direction to explain the phenomenon described in [19] (see Remark 3), and to exploit regularization properties of some algorithms (see Section 4).

## 1.3 Further related literature

In [20], the mutual information between the input and the output of binary classification learning algorithms is used to obtain high probability generalization bounds.

PAC-Bayesian bounds are another type of algorithm-dependent bounds which are concerned with finding high probability generalization bounds for randomized classifiers [5]. These bounds define a hierarchy over the hypothesis set by using a prior distribution on that set [4]. As discussed in [20], there is a connection and similarity between PAC-Bayesian bounds and the mutual information bound, both using the variational representation of relative entropy in their proofs. In [21] and [22], the authors combine the ideas of PAC-Bayesian bounds with generic chaining and create high probability bounds for randomized classifiers. Their use of an auxiliary sample set and the notion of average distance between partitions makes their bounds conceptually different from our work. However, their bounds have the advantage to exploit the variance of the hypotheses and to give high probability results.

In the probability theory literature, Fernique [23] gives upper and lower bounds on the expected bias of an algorithm (or a selection rule) which chooses its output from a Gaussian process, by using a

chaining argument while taking into account the marginal distribution of the algorithm output. We further utilize the dependence between the algorithm input and output and the stochasticity of the algorithm, and we give results for more general processes. However, we only obtain upper bounds in this paper.

## 1.4 Notation

In the framework of supervised statistical learning, $\mathcal{X}$ is the instances domain, $\mathcal{Y}$ is the labels domain and $\mathsf{Z} = \mathcal{X} \times \mathcal{Y}$ denotes the examples domain. Furthermore, $\mathcal{H} = \{h_w : w \in \mathcal{W}\}$ is the hypothesis set where the hypotheses are indexed by an index set $\mathcal{W}$, and there is a nonnegative loss function $\ell : \mathcal{H} \times \mathsf{Z} \to \mathbb{R}^+$. A learning algorithm receives the training set $S = (Z_1, Z_2, ..., Z_n)$ of $n$ examples with i.i.d. random elements drawn from $\mathsf{Z}$ with distribution $\mu$. Then it picks an element $h_W \in \mathcal{H}$ as the output hypothesis according to a random transformation $P_{W|S}$ (thus, we are allowing randomized algorithms). For any $w \in \mathcal{W}$, let

$$L_\mu(w) \triangleq \mathbb{E}[\ell(h_w, Z)], \quad Z \sim \mu \tag{8}$$

denote the statistical (or population) risk of hypothesis $h_w$. For a given training set $S$, the empirical risk of hypothesis $h_w$ is defined as

$$L_S(w) \triangleq \frac{1}{n} \sum_{i=1}^n \ell(h_w, Z_i), \tag{9}$$

and the generalization error of hypothesis $h_w$ (dependent on the training set) is defined as

$$\text{gen}(w) \triangleq L_\mu(w) - L_S(w). \tag{10}$$

Averaging with respect to the joint distribution $P_{S,W} = \mu^{\otimes n} P_{W|S}$, we denote the expected generalization error and the expected absolute value of generalization error by

$$\text{gen}(\mu, P_{W|S}) \triangleq \mathbb{E}[L_\mu(W) - L_S(W)], \tag{11}$$

and

$$\text{gen}^+(\mu, P_{W|S}) \triangleq \mathbb{E}[|L_\mu(W) - L_S(W)|], \tag{12}$$

respectively. Our purpose is to find upper bounds on $\text{gen}(\mu, P_{W|S})$ and $\text{gen}^+(\mu, P_{W|S})$.

Let $X_\mathcal{N} \triangleq \{X_i : i \in \mathcal{N}\}$ denote a random process indexed by the elements of the set $\mathcal{N}$. Let $\mathbf{0}$ denote the identically zero function. In this paper, all logarithms are in natural base and all information theoretic measures are in nats. $H(X)$ denotes the Shannon entropy of a discrete random variable $X$, and $h(Y)$ denotes the differential entropy of an absolutely continuous random variable $Y$.

## 2 Main results

Assume that $\{X_t\}_{t \in T}$ is a random process with index set $T$. In the chaining method, we impose a metric $d$ on $T$ which describes the dependencies between the random variables. The widely used *subgaussian processes* capture this notion and they arise in many applications:

**Definition 1** (Subgaussian process). *The random process $\{X_t\}_{t \in T}$ on the metric space $(T, d)$ is called* subgaussian *if $\mathbb{E}[X_t] = 0$ for all $t \in T$ and*

$$\mathbb{E}\left[e^{\lambda(X_t - X_s)}\right] \leq e^{\frac{1}{2}\lambda^2 d^2(t,s)} \quad \text{for all} \quad t, s \in T, \lambda \geq 0. \tag{13}$$

For example, based on the Azuma–Hoeffding inequality, $\{\text{gen}(w)\}_{w \in \mathcal{W}}$ is a subgaussian process with the metric

$$d(\text{gen}(w), \text{gen}(v)) \triangleq \frac{\|\ell(h_w, \cdot) - \ell(h_v, \cdot)\|_\infty}{\sqrt{n}}, \tag{14}$$

regardless of the choice of distribution $\mu$ on $\mathsf{Z}$.

The following is a technical assumption which holds in almost all cases of interest:

**Definition 2** (Separable process). *The random process $\{X_t\}_{t \in T}$ is called* separable *if there is a countable set $T_0 \subseteq T$ such that $X_t \in \lim_{\substack{s \to t \\ s \in T_0}} X_s$ for all $t \in T$ a.s., where $x \in \lim_{\substack{s \to t \\ s \in T_0}} x_s$ means that there is a sequence $(s_n)$ in $T_0$ such that $s_n \to t$ and $x_{s_n} \to x$.*

For example, if $t \to X_t$ is continuous a.s., then $X_t$ is a separable process [9].

Our main results rely on the notion of increasing sequence of $\epsilon$-partitions of the metric space $(T, d)$:

**Definition 3** (Increasing sequence of $\epsilon$-partitions)**.** *We call a partition $\mathcal{P} = \{A_1, A_2, ..., A_m\}$ of the set $T$ an $\epsilon$-partition of the metric space $(T, d)$ if for all $i = 1, 2, ..., m$, $A_i$ can be contained within a ball of radius $\epsilon$. A sequence of partitions $\{\mathcal{P}_k\}_{k=m}^{\infty}$ of a set $T$ is called an* increasing sequence *if for all $k \geq m$ and each $A \in \mathcal{P}_{k+1}$, there exists $B \in \mathcal{P}_k$ such that $A \subseteq B$. For any such sequence and any $t \in T$, let $[t]_k$ denote the unique set $A \in \mathcal{P}_k$ such that $t \in A$.*

Assume now that $(T, d)$ is a bounded metric space, and let $k_1(T)$ be an integer such that $2^{-(k_1(T)-1)} \geq \operatorname{diam}(T)$. We have the following upper bounds on $\operatorname{gen}(\mu, P_{W|S})$ and $\operatorname{gen}^+(\mu, P_{W|S})$ based on the mutual information between the training set $S$ and the discretized output of the learning algorithm, where each of these mutual information terms is multiplied by an exponentially decreasing weight $2^{-k}$, in which the exponent measures how finely the output $W$ of the learning algorithm is discretized:

**Theorem 3.** *Assume that $\{\operatorname{gen}(w)\}_{w \in \mathcal{W}}$ is a separable subgaussian process on the bounded metric space $(\mathcal{W}, d)$. Let $\{\mathcal{P}_k\}_{k=k_1(\mathcal{W})}^{\infty}$ be an increasing sequence of partitions of $\mathcal{W}$, where for each $k \geq k_1(\mathcal{W})$, $\mathcal{P}_k$ is a $2^{-k}$-partition of $(\mathcal{W}, d)$.*

*(a)*

$$\operatorname{gen}(\mu, P_{W|S}) \leq 3\sqrt{2} \sum_{k=k_1(\mathcal{W})}^{\infty} 2^{-k} \sqrt{I([W]_k; S)}, \tag{15}$$

*(b) If $\mathbf{0} \in \{\ell(h_w, \cdot) : w \in \mathcal{W}\}$, then*

$$\operatorname{gen}^+(\mu, P_{W|S}) \leq 3\sqrt{2} \sum_{k=k_1(\mathcal{W})}^{\infty} 2^{-k} \sqrt{I([W]_k; S) + \log 2}. \tag{16}$$

**Remark 1.** Based on the general definition of mutual information with partitions ([24, p. 252]), we have $I(W; S) = \sup_k I([W]_k; S)$ therefore $I([W]_k; S) \to I(W; S)$ as $k \to \infty$.

Theorem 3 is stated in the context of statistical learning. The more general counterpart in the context of random processes is:

**Theorem 4.** *Assume that $\{X_t\}_{t \in T}$ is a separable subgaussian process on the bounded metric space $(T, d)$. Let $\{\mathcal{P}_k\}_{k=k_1(T)}^{\infty}$ be an increasing sequence of partitions of $T$, where for each $k \geq k_1(T)$, $\mathcal{P}_k$ is a $2^{-k}$-partition of $(T, d)$.*

*(a)*

$$\mathbb{E}[X_W] \leq 3\sqrt{2} \sum_{k=k_1(T)}^{\infty} 2^{-k} \sqrt{I([W]_k; X_T)}. \tag{17}$$

*(b) For any arbitrary $t_0 \in T$,*

$$\mathbb{E}[|X_W - X_{t_0}|] \leq 3\sqrt{2} \sum_{k=k_1(T)}^{\infty} 2^{-k} \sqrt{I([W]_k; X_T) + \log 2}. \tag{18}$$

Note that in Theorem 4 if we let $T \triangleq \mathcal{W}$ and $X_w \triangleq \operatorname{gen}(w)$ for all $w \in \mathcal{W}$, then for each $k \geq k_1(T)$, due to the Markov chain

$$X_T = \{\operatorname{gen}(w)\}_{w \in \mathcal{W}} \leftrightarrow S \leftrightarrow W \leftrightarrow [W]_k, \tag{19}$$

and the data processing inequality, we have $I([W]_k; X_T) \leq I([W]_k; S)$. Therefore Theorem 3 follows from Theorem 4. The proof of Theorem 4 and the etymology of "chaining mutual information" is given in Section 3.

**Remark 2.** For random processes other than subgaussian processes, where the tail of increments are controlled by a function $\psi$, similar results can be derived from Theorem 12 in Section D of the supplementary material.

Both Theorem 3 and Theorem 4 capture the dependencies between the hypotheses by utilizing a metric $d$, and they are algorithm-dependent as the mutual information between the algorithm's discretized output and its input appears in their bounds. Now, to demonstrate the power of Theorem 4 and to compare it with the existing results in the literature, consider the following example:

**Example 1.** Let $T$ be an arbitrary subset of $\mathbb{R}^n$, and $G^n \triangleq (G_1, ..., G_n) \sim \mathcal{N}(0, I_n)$ be a standard normal random vector in $\mathbb{R}^n$. The *canonical Gaussian process* is defined as $\{X_t\}_{t \in T}$, where

$$X_t \triangleq \langle t, G^n \rangle \quad \text{for all } t \in T. \tag{20}$$

Note that $\{X_t\}_{t \in T}$ is a subgaussian process on the metric space $(T, d)$, where $d$ is the Euclidean distance.

Consider a canonical Gaussian process where $n = 2$ and $T \triangleq \{t \in \mathbb{R}^2 : \|t\|_2 = 1\}$. The process $\{X_t\}_{t \in T}$ can be reparameterized according to the phase of each point $t \in T$: the random variable $X_t$ can also be denoted as $X_\phi$, where $\phi \in [0, 2\pi)$ is the phase of $t$. In other words, $\phi$ is the unique number in $[0, 2\pi)$ such that $t = (\sin \phi, \cos \phi)$. Henceforth, we will assume the indices are in the phase form.

Let the relation between the input $X_T$ of an algorithm and its output $W$ be as

$$W \triangleq \left( \text{argmax}_{\phi \in [0, 2\pi)} X_\phi \right) \oplus Z \pmod{2\pi}, \tag{21}$$

where the noise $Z$ is independent from $X_T$, and has an atom with probability mass $\epsilon$ on 0, and $1 - \epsilon$ probability is uniformly distributed on $(-\pi, \pi)$. Note that since $Z$ has a singular (degenerate) part, $h(Z) = -\infty$.

Due to symmetry, $W$ has uniform distribution over $[0, 2\pi)$. But we have

$$I(W; X_T) = h(W) - h(W|X_T) \tag{22}$$

$$= \log 2\pi - h\left( \text{argmax}_{\phi \in [0, 2\pi)} X_\phi \oplus Z \Big| X_T \right) \tag{23}$$

$$= \log 2\pi - h(Z|X_T) \tag{24}$$

$$= \log 2\pi - h(Z) \tag{25}$$

$$= \infty. \tag{26}$$

Hence the upper bound on $\mathbb{E}[X_W]$ due to the mutual information method (Theorem 2) blows up:

$$\mathbb{E}[X_W] \leq \sqrt{2I(W; X_T)} = \infty. \tag{27}$$

Note that $2^{-(-2)} \geq \text{diam}(T) = 2$. Therefore let $k_1(T) \leftarrow -1$ and for all integers $k \geq -1$, define

$$\mathcal{P}_k \triangleq \left\{ \left[ 0, \frac{2\pi}{2^{k+2}} \right), \left[ \frac{2\pi}{2^{k+2}}, 2 \times \frac{2\pi}{2^{k+2}} \right), ..., \left[ (2^{k+2} - 1) \frac{2\pi}{2^{k+2}}, 2\pi \right) \right\}. \tag{28}$$

It is clear that $\{\mathcal{P}_k\}_{k=-1}^{\infty}$ is an increasing sequence of partitions of $T$. Furthermore, for each $k \geq -1$, the length of the arc of each set in $\mathcal{P}_k$ is $\delta_k \triangleq \frac{2\pi}{2^{k+2}} < 2^{1-k}$. Thus each $\mathcal{P}_k$ is a $2^{-k}$-partition of $(T, d)$ and $|\mathcal{P}_k| = 2^{k+2}$ (see Figure 1).

Now by using the classical chaining method (Theorem 1) to upper bound $\mathbb{E}[X_W]$ by upper bounding $\mathbb{E}[\sup_{\phi \in [0, 2\pi)} X_\phi]$ and ignoring the algorithm, we get

$$\mathbb{E}[X_W] \leq \mathbb{E}\left[ \sup_{\phi \in [0, 2\pi)} X_\phi \right] \tag{29}$$

$$\leq 3\sqrt{2} \sum_{k=-1}^{\infty} 2^{-k} \sqrt{\log 2^{k+2}} \tag{30}$$

$$= 19.0352...[3] \tag{31}$$

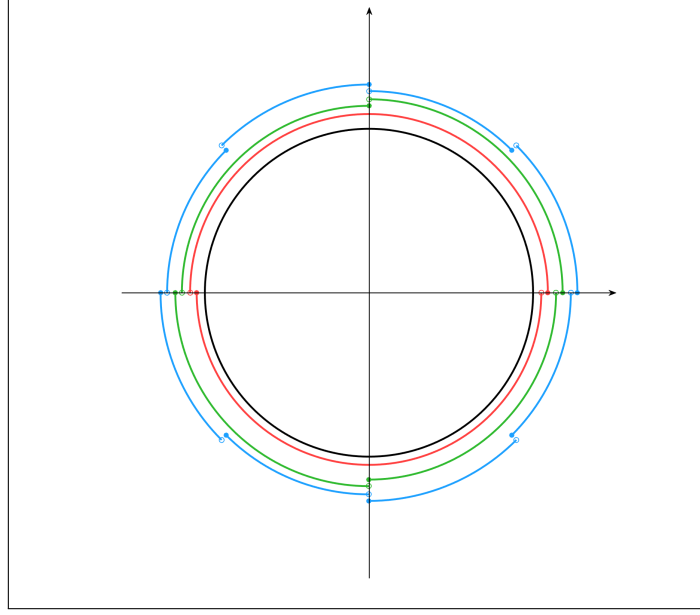

Figure 1: Depiction of $T, \mathcal{P}_{-1}, \mathcal{P}_0$ and $\mathcal{P}_1$ in the $\mathbb{R}^2$ plane. (The three partitions are magnified for clarity.)

On the other hand, for every $k \geq -1$ we have

$$I([W]_k; X_T) = H([W]_k) - H([W]_k|X_T) \tag{32}$$

$$= \log 2^{k+2} - H\left(\left[\left(\operatorname{argmax}_{\phi \in [0, 2\pi)} X_\phi\right) \oplus Z\right]_k \bigg| X_T\right) \tag{33}$$

$$= \log 2^{k+2} - H\left(\epsilon + \frac{1-\epsilon}{2^{k+2}}, \frac{1-\epsilon}{2^{k+2}}, \dots, \frac{1-\epsilon}{2^{k+2}}\right). \tag{34}$$

Therefore, based on the chained mutual information method (Theorem 4), we have

$$\mathbb{E}[X_W] \leq 3\sqrt{2} \sum_{k=-1}^{\infty} 2^{-k} \sqrt{I([W]_k; X_T)} \tag{35}$$

$$= 3\sqrt{2} \sum_{k=-1}^{\infty} 2^{-k} \sqrt{\log 2^{k+2} - H\left(\epsilon + \frac{1-\epsilon}{2^{k+2}}, \frac{1-\epsilon}{2^{k+2}}, \dots, \frac{1-\epsilon}{2^{k+2}}\right)} \tag{36}$$

Numerical values of the right side of (36) for different values of $\epsilon$ are given in Table 1 (CMI bound). Note that indeed $I([W]_k; X_T) \to I(W; X_T) = \infty$ as $k \to \infty$. However, the slow rate of that convergence and the existence of the $2^{-k}$ term makes the sum not only finite, but very small. In fact, as $\epsilon \to 0$, the right side of (36) tends to 0 as well.

It is interesting to notice that for this toy example, the exact values of $\mathbb{E}\left[\sup_{\phi \in [0, 2\pi)} X_\phi\right]$ and $\mathbb{E}[X_W]$ can be computed. As $\sup_{\phi \in [0, 2\pi)} X_\phi$ has a Rayleigh distribution, we have $\mathbb{E}\left[\sup_{\phi \in [0, 2\pi)} X_\phi\right] = \sqrt{\frac{\pi}{2}} = 1.253...$ . Since the noise $Z$ is independent from $X_T$, the effect of its continuous part cancels out, and we have $\mathbb{E}[X_W] = \epsilon\sqrt{\frac{\pi}{2}}$. See Table 1.

Table 1: $\mathbb{E}[X_W]$ and its upper bounds

| $\epsilon$ | $\frac{1}{20}$ | $\frac{1}{30}$ | $\frac{1}{40}$ | $\frac{1}{50}$ | $\frac{1}{100}$ | $\frac{1}{200}$ | $\frac{1}{400}$ |
|---|---|---|---|---|---|---|---|
| $2\sqrt{I(W;X_T)}$ | $\infty$ | $\infty$ | $\infty$ | $\infty$ | $\infty$ | $\infty$ | $\infty$ |
| Chaining bound | 19.0352 | 19.0352 | 19.0352 | 19.0352 | 19.0352 | 19.0352 | 19.0352 |
| CMI bound | 1.1013 | 0.7507 | 0.5709 | 0.4612 | 0.2364 | 0.1204 | 0.0610 |
| $\mathbb{E}[X_W]$ | 0.0626 | 0.0417 | 0.0313 | 0.0250 | 0.0125 | 0.0062 | 0.0031 |

**Remark 3.** Notice that in Example 1 there exists an independent additive noise term $Z$ which has a degenerate part, causing the mutual information bound to blow up. Similarly, as discussed in [19], the mutual information bound for perturbed SGD or any iterative algorithm which adds degenerate noise in each iteration blows up. Example 1 illustrates that combining the mutual information method with the chaining method as in our bound could give tight generalization bounds for such algorithms as well.

**Remark 4.** It is clear that having degenerate noise is not necessary to observe that the chained mutual information bound is tighter than the mutual information bound; this is just an extreme case for which the mutual information bound blows up. For instance, in Example 1, one can replace $Z$ with a sequence of continuous random variables which converge to $Z$ in distribution.

## 3   Proof outline

Here we provide an outline of the proof of Theorem 4. As noted in Section 2, Theorem 3 follows from Theorem 4.

For an arbitrary $k \geq k_1(T)$, consider $\mathcal{P}_k = \{A_1, A_2, ..., A_m\}$. Since $\mathcal{P}_k$ is a $2^{-k}$-partition of $(T, d)$, by definition there exists a set (or a multiset) $\mathcal{N}_k \triangleq \{a_1, a_2, ..., a_m\} \subseteq T$ and a mapping $\pi_{\mathcal{N}_k} : T \rightarrow \mathcal{N}_k$ such that $\pi_{\mathcal{N}_k}(t) = a_i$ if $t \in A_i$, and further $d(t, \pi_{\mathcal{N}_k}(t)) \leq 2^{-k}$, for all $i = 1, 2, ..., m$. Therefore $\mathcal{N}_k$ is a $2^{-k}$-net and $\pi_{\mathcal{N}_k}$ is its associated mapping. It is also clear that for an arbitrary $t_0 \in T$, $\mathcal{N}_{k_0} \triangleq \{t_0\}$ is a $2^{-(k_1(T)-1)}$-net. Note that for any integer $n \geq k_1(T)$ we can write

$$X_W = X_{t_0} + \sum_{k=k_1(T)}^{n} \left( X_{\pi_{\mathcal{N}_k}(W)} - X_{\pi_{\mathcal{N}_{k-1}}(W)} \right) + \left( X_W - X_{\pi_{\mathcal{N}_n}(W)} \right). \tag{37}$$

Since by the definition of subgaussian processes the process is centered, we have $\mathbb{E}[X_{t_0}] = 0$. Thus

$$\mathbb{E}[X_W] - \mathbb{E}\left[X_W - X_{\pi_{\mathcal{N}_n}(W)}\right] = \sum_{k=k_1(T)}^{n} \mathbb{E}\left[X_{\pi_{\mathcal{N}_k}(W)} - X_{\pi_{\mathcal{N}_{k-1}}(W)}\right]. \tag{38}$$

For every $k \geq k_1(T)$, $\{X_{\pi_{\mathcal{N}_k}(t)} - X_{\pi_{\mathcal{N}_{k-1}}(t)}\}_{t \in T}$ is a subgaussian process with at most $|\mathcal{N}_k||\mathcal{N}_{k-1}|$ distinct terms, hence a finite process. Based on the triangle inequality,

$$d\left(\pi_{\mathcal{N}_k}(t), \pi_{\mathcal{N}_{k-1}}(t)\right) \leq d\left(t, \pi_{\mathcal{N}_k}(t)\right) + d\left(t, \pi_{\mathcal{N}_{k-1}}(t)\right)$$
$$\leq 3 \times 2^{-k}. \tag{39}$$

Note that knowing the value of $\left(\pi_{\mathcal{N}_k}(W), \pi_{\mathcal{N}_{k-1}}(W)\right)$ is enough to determine which one of the random variables $\left\{X_{\pi_{\mathcal{N}_k}(t)} - X_{\pi_{\mathcal{N}_{k-1}}(t)}\right\}_{t \in T}$ is chosen according to $W$. Therefore $\left(\pi_{\mathcal{N}_k}(W), \pi_{\mathcal{N}_{k-1}}(W)\right)$ is playing the role of the random index, and since $X_{\pi_{\mathcal{N}_k}(t)} - X_{\pi_{\mathcal{N}_{k-1}}(t)}$ is $d^2\left(\pi_{\mathcal{N}_k}(t), \pi_{\mathcal{N}_{k-1}}(t)\right)$-subgaussian, based on Theorem 2, an application of data processing inequality and by summation, we obtain

$$\sum_{k=k_1(T)}^{n} \mathbb{E}\left[X_{\pi_{\mathcal{N}_k}(W)} - X_{\pi_{\mathcal{N}_{k-1}}(W)}\right] \leq \sum_{k=k_1(T)}^{n} 3\sqrt{2} \times 2^{-k} \sqrt{I(\pi_{\mathcal{N}_k}(W), \pi_{\mathcal{N}_{k-1}}(W); X_T)}. \tag{40}$$

Notice the chain of mutual information terms in the right side of (40). Since $\{\mathcal{P}_k\}_{k=k_1(T)}^{\infty}$ is an increasing sequence of partitions, for any $t \in T$, knowing $\mathcal{N}_k(t)$ will uniquely determine $\mathcal{N}_{k-1}(t)$. Therefore

$$I\left(\pi_{\mathcal{N}_k}(W), \pi_{\mathcal{N}_{k-1}}(W); X_T\right) = I\left(\pi_{\mathcal{N}_k}(W); X_T\right) \tag{41}$$
$$= I\left([W]_k; X_T\right). \tag{42}$$

The rest of the proof follows from the definition of separable processes (Definition 2). For more details, see proof of Theorem 11 in Section D of the supplementary material.

## 4   Additional result: small subset property

We adjusted the conservative chaining method in random processes theory to learning problems by taking into account information about the algorithm, with the chained mutual information method. In this section, we state a result in which such information could make the bounds much tighter.

It is known that for linear models, the stochastic gradient descent (SGD) algorithm always converges to a solution with small norm [1]. Inspired by this observation, we tighten Dudley's inequality (Theorem 1), given the following regularization property: the output $W$ of an algorithm, with high probability, chooses a hypothesis from a subset of the hypothesis set with small covering numbers:

**Theorem 5** (Small subset property). *Assume that $\{X_t\}_{t \in T}$ is a separable subgaussian process on the bounded metric space $(T, d)$. Let $\{T_1, T_2\}$ be a partition of $T$ and assume that $W$ is a random variable taking values on $T$ with $\mathbb{P}[W \in T_1] = \alpha$. Then we have*

$$\mathbb{E}[X_W] \leq 6 \sum_{k=k_1(T)}^{\infty} 2^{-k} \sqrt{\alpha \log N(T_1, d, 2^{-k}) + (1-\alpha) \log N(T_2, d, 2^{-k}) + H(\alpha)}. \tag{43}$$

Proof of Theorem 5 appears in Section D of the supplementary material. Note that the right side of (43) becomes much smaller than Dudley's bound when $\alpha$ is close to 1 and the covering numbers of $T_1$ (the small subset) are much smaller than the covering numbers of $T_2$.

**Remark 5.** One can upper bound the right side of (43) by replacing $N(T_2, d, 2^{-k})$ with $N(T, d, 2^{-k})$. This is particularly useful when bounding the latter is easier than the former.

## 5   Conclusion

We combined ideas from information theory and from high dimensional probability to obtain a generalization bound that takes into account both the dependencies between the hypotheses and the dependence between the input and the output of a learning algorithm. We showed on an example that our chained mutual information bound significantly outperforms previous bounds and gets close to the true generalization error. Under a natural regularization property of the learning algorithm, we provided a corollary of our bound which tightens Dudley's inequality; i.e. when the learning algorithm chooses its output from a small subset of hypotheses with high probability.

## 6   Acknowledgments

We gratefully acknowledge discussions with Ramon van Handel on the topic of chaining. This work was partly supported by the NSF CAREER Award CCF-1552131.

## Footnotes

[2]Here we are referring to the Rademacher average of the entire hypothesis set. There exist other notions of Rademacher averages which are used in algorithm-dependent bounds, such as in local Rademacher complexities [15].

[3]The exact value of the bound in Theorem 1 is slightly smaller, since with our partitions we are using a rough approximate for the covering numbers. For example, at scale $2^{-(-1)}$, the covering number is 1, while we have used partition $\mathcal{P}_{-1}$ with $|\mathcal{P}_{-1}| = 2$ sets.

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
