[Supplementary Material · Chaining Mutual Information_ NeurIPS 2018_Supplementary.pdf]

In section B which deals with finite random processes and which serves as the basic foundation of chaining, the known results of maximal inequality (Proposition 1) and its improvement via mutual information (Theorem 7) are reviewed. Then we give a condition for a random process in Corollary 3, for which the result of Theorem 7 can be improved by upper *and* lower bounding $\mathbb{E}[X_W]$. The aforementioned results concern $\mathbb{E}[X_W]$; in Theorem 8 we obtain inequalities for the tail behavior of $X_W$.

In the next step of building upon the results of section B, to be able to handle infinite processes, in section C we introduce the notion of $\epsilon$-*nets* (see Definition 8) and its related definitions, and in Theorem 9 we upper bound $\mathbb{E}[X_W]$ for *Lipschitz processes* (see Definition 7) using mutual information. This is the strengthened version of the so-called $\epsilon$-net argument, with the usage of mutual information. Remark 9 discusses upper bounding $|\mathbb{E}[X_W]|$ for Lipschitz processes.

In the last step, in section D, we loosen the "almost sure" Lipschitz condition of the dependencies of the random variables of a process to a "in probability" condition, defined as subgaussian processes (see Definition 9). After reviewing the classical chaining result of Dudley's inequality (Theorem 10), we combine the mutual information method and the chaining method in Theorem 11 for subgaussian processes, and in Thoerem 12 for more general processes.

## A  Preliminaries

**Definition 4** (Cumulant generating function). *Let $X$ be a real valued random variable. The* cumulant generating function *of $X$ is defined as $\Lambda_X(\lambda) \triangleq \log \mathbb{E}[e^{\lambda X}]$ for all $\lambda \in \mathbb{R}$.*

The following lemma is a well known fact about the cumulant generating function:

**Lemma 1.** *Let $X$ be a random variable. Then its cumulant generating function $\Lambda_X$ is convex, $\Lambda_X(0) = 0$ and $\Lambda'_X(0) = \mathbb{E}[X]$.*

An important and widely used class of random variables is the class of subgaussian random variables:

**Definition 5** (Subgaussian random variables). *The random variable $X$ is called $\sigma^2$-subgaussian if $\mathbb{E}[e^{\lambda(X-\mathbb{E}X)}] \leq e^{\frac{\lambda^2\sigma^2}{2}}$ for all $\lambda \in \mathbb{R}$. In particular, if $X$ is $\sigma^2$-subgaussian and $\mathbb{E}[X] = 0$, then its cumulant generating function satisfies $\Lambda_X(\lambda) \leq \frac{\lambda^2\sigma^2}{2}$ for all $\lambda \in \mathbb{R}$. The constant $\sigma^2$ is called the* variance proxy.

We will use the notion of Legendre dual, defined as follows, in our bounds.

**Definition 6** (Legendre dual). *For a convex function $\psi : \mathbb{R}_+ \to \mathbb{R}$, the* Legendre dual $\psi^* : \mathbb{R} \to \mathbb{R}$ *is defined as*

$$\psi^*(x) \triangleq \sup_{\lambda \geq 0}\{\lambda x - \psi(\lambda)\} \ \text{ for all } \ x \in \mathbb{R}. \tag{44}$$

For a proof of the next lemma see [9, p. 115]:

**Lemma 2** (Legendre dual properties). *Let $\psi : \mathbb{R}_+ \to \mathbb{R}$ be a convex function and $\psi(0) = \psi'(0) = 0$. Then $\psi^*(x)$ is a convex, strictly increasing, nonnegative and unbounded function for $x \geq 0$, and $\psi^*(0) = 0$. Therefore its inverse $\psi^{*-1}(y)$ is well defined for $y \geq 0$.*

From Definition 5, if $X$ is $\sigma^2$-subgaussian and $\mathbb{E}[X] = 0$ then $\Lambda_{X_t}(\lambda) \leq \frac{\lambda^2\sigma^2}{2}$. The following lemma gives the Legendre dual inverse of $\psi(\lambda) \triangleq \frac{\lambda^2\sigma^2}{2}$.

**Lemma 3.** *Let $\psi(\lambda) \triangleq \frac{\lambda^2\sigma^2}{2}$ for all $\lambda \geq 0$. Then $\psi^{*-1}(x) = \sqrt{2\sigma^2 x}$ for all $x \in \mathbb{R}$.*

The following is the well-known Chernoff bound:

**Lemma 4** (Chernoff). *Let $X$ be a random variable, and $\psi$ be a function such that $\Lambda_X(\lambda) \leq \psi(\lambda)$ for all $\lambda \geq 0$. Then*

$$\mathbb{P}[X \geq x] \leq e^{-\psi^*(x)} \ \text{ for all } \ x \in \mathbb{R}. \tag{45}$$

The variational representation of relative entropy is a useful information theoretic tool:

**Theorem 6** (Variational representation of relative entropy)**.** *Let $X$ and $Y$ be random variables taking values on $\mathcal{A}$ with distributions $P_X$ and $P_Y$, respectively. Then*

$$D(P_X \| P_Y) = \max_{f \in \mathsf{F}} \left\{ \mathbb{E}\left[f(X)\right] - \log \mathbb{E}\left[e^{f(Y)}\right] \right\}, \tag{46}$$

*where the maximum is with respect to* $\mathsf{F} = \left\{ f : \mathcal{A} \to \mathbb{R} \text{ s.t. } \mathbb{E}[e^{f(Y)}] < \infty \right\}$, *and is achieved by* $f^*(a) = \imath_{X \| Y}(a)$.

## B Finite processes (random vectors)

In this section we consider a random process $\{X_t\}_{t \in T}$ where $T$ is a finite set. The following is a well known result (see [12, Theorem 2.5]):

**Proposition 1** (Maximal inequality)**.** *Let $\{X_t\}_{t \in T}$ be a random process and $T$ a finite set. Assume that $\Lambda_{X_t}(\lambda) \leq \psi(\lambda)$ for all $\lambda \geq 0$ and $t \in T$, where $\psi$ is convex and $\psi(0) = \psi'(0) = 0$. Then*

$$\mathbb{E}\left[\sup_{t \in T} X_t\right] \leq \psi^{*-1}(\log |T|). \tag{47}$$

*In particular, if $X_t$ is $\sigma^2$-subgaussian and $\mathbb{E}[X_t] = 0$ for every $t \in T$, then*

$$\mathbb{E}\left[\sup_{t \in T} X_t\right] \leq \sqrt{2\sigma^2 \log |T|}. \tag{48}$$

**Remark 6.** Note that based on Lemma 1, for all $t \in T$, the condition $\Lambda_{X_t}(\lambda) \leq \psi(\lambda)$ for all $\lambda \geq 0$ and $\psi'(0) = 0$ implies that $\mathbb{E}[X_t] = 0$.

**Proposition 2.** *If in addition to the assumptions of Proposition 1, we assume that $\Lambda_{X_t}(-\lambda) \leq \psi(\lambda)$ for all $\lambda \geq 0$ and $t \in T$, then we have*

$$\mathbb{E}\left[\sup_{t \in T} |X_t|\right] \leq \psi^{*-1}\left(\log(2|T|)\right). \tag{49}$$

*In particular, if $X_t$ is $\sigma^2$-subgaussian and $\mathbb{E}[X_t] = 0$ for every $t \in T$, then*

$$\mathbb{E}\left[\sup_{t \in T} |X_t|\right] \leq \sqrt{2\sigma^2 \log (2|T|)}. \tag{50}$$

*Proof.* Apply Proposition 1 on the random process $\{X_t\}_{t \in T} \cup \{-X_t\}_{t \in T}$. $\qquad \square$

The next result bounds $\mathbb{E}[X_W]$, where $W$ is a random variable taking values on $T$:

**Theorem 7.** *[8], [16] Let $\{X_t\}_{t \in T}$ be a random process and $T$ a finite set. Assume that $\Lambda_{X_t}(\lambda) \leq \psi(\lambda)$ for all $\lambda \geq 0$ and $t \in T$, where $\psi$ is convex and $\psi(0) = \psi'(0) = 0$, and let $W$ be a random variable taking values on $T$. Then*

$$\mathbb{E}[X_W] \leq \psi^{*-1}(I(W; X_T)). \tag{51}$$

*In particular, if $X_t$ is $\sigma^2$-subgaussian and $\mathbb{E}[X_t] = 0$ for every $t \in T$, then*

$$\mathbb{E}[X_W] \leq \sqrt{2\sigma^2 I(W; X_T)}. \tag{52}$$

Based on Lemma 2, $\psi^{*-1}$ is an increasing function. Therefore one can replace $I(W; X_T)$ with any larger quantity in the right side of (51). For example,

$$\mathbb{E}[X_W] \leq \psi^{*-1}(I(W; X_T))$$
$$\leq \psi^{*-1}(H(W)). \tag{53}$$

Since $W$ takes values on $T$, we have $H(W) \leq \log |T|$. Therefore the right side of (51) is not larger than the right side of (47).

Based on Lemma 2, the right side of (51) is zero if and only if $I(W; X_T) = 0$, i.e. $W$ is independent of $X_T$. In this case, (51) turns into an equality: based on Remark 6 we have $\mathbb{E}[X_t] = 0$ for all $t \in T$, hence $\mathbb{E}[X_W] = \mathbb{E}[\mathbb{E}[X_W | W]] = 0$.

Now, by adding an assumption, we prove upper *and* lower bounds for $\mathbb{E}[X_W]$, and an upper bound for $\mathbb{E}[|X_W|]$. We should mention that the proof of part (b) of the following proposition is similar to the proof of Theorem 4 in [18].

**Proposition 3.** *If in addition to the assumptions of Theorem 7, we assume that $\Lambda_{X_t}(-\lambda) \leq \psi(\lambda)$ for all $\lambda \geq 0$ and $t \in T$, then we have*

*(a)*
$$|\mathbb{E}[X_W]| \leq \psi^{*-1}(I(W; X_T)), \tag{54}$$

*(b)*
$$\mathbb{E}[|X_W|] \leq \psi^{*-1}\left(I(W; X_T) + \log 2\right). \tag{55}$$

*Proof.*

(a) Apply Theorem 7 to the process $\{-X_t\}_{t \in T}$, while noting that $\Lambda_{-X_t}(\lambda) = \Lambda_{X_t}(-\lambda)$ for all $\lambda \geq 0$ and $t \in T$, and $I(W; -X_T) = I(W; X_T)$, since mutual information is invariant to one-to-one functions.

(b) Define the random process $\overline{X} = \{X_{t,w}\}_{\substack{t \in T \\ w \in \{0,1\}}}$ such that

$$X_{t,w} \triangleq \begin{cases} X_t & t \in T, w = 0 \\ -X_t & t \in T, w = 1 \end{cases}$$

and let $R$ be a random variable taking values on $\{0, 1\}$ such that

$$R = \begin{cases} 0 & \text{if } X_W \geq 0 \\ 1 & \text{if } X_W < 0 \end{cases}.$$

Based on Theorem 7 applied on the random process $\overline{X}$ and random variables $W$ and $R$, and based on the chain rule of entropy, we get

$$\mathbb{E}[|X_W|] = \mathbb{E}[X_{W,R}] \tag{56}$$
$$\leq \psi^{*-1}(I(W, R; \overline{X})) \tag{57}$$
$$= \psi^{*-1}(H(W, R) - H(W, R|\overline{X})) \tag{58}$$
$$= \psi^{*-1}((H(W) + H(R|W)) - (H(W|\overline{X}) + H(R|W, \overline{X}))) \tag{59}$$
$$= \psi^{*-1}((H(W) + H(R|W)) - H(W|\overline{X})) \tag{60}$$
$$= \psi^{*-1}(H(W) - H(W|X_T) + H(R|W)) \tag{61}$$
$$= \psi^{*-1}(I(W; X_T) + H(R|W)) \tag{62}$$
$$\leq \psi^{*-1}(I(W; X_T) + H(R)) \tag{63}$$
$$\leq \psi^{*-1}(I(W; X_T) + \log 2). \tag{64}$$

$\square$

**Corollary 1.** *If $T$ is a finite set, $\psi(\lambda) \triangleq \frac{\lambda^2 \sigma^2}{2}$, and for all $t \in T$, $X_t$ is $\sigma^2$-subgaussian and $\mathbb{E}[X_t] = 0$, then the conditions of Theorem 3 is satisfied, and (52) can be improved to*

$$|\mathbb{E}[X_W]| \leq \sqrt{2\sigma^2 I(W; X_T)}, \tag{65}$$

*as was shown in [8].*

The previous results concerned $\mathbb{E}[\sup_{t \in T} X_t]$ and $\mathbb{E}[X_W]$. We now state a result for estimating the tail probability of $\sup_{t \in T} X_t$:

**Proposition 4.** *[9] Let $\{X_t\}_{t \in T}$ be a random process and $T$ a finite set. Assume that $\Lambda_{X_t}(\lambda) \leq \psi(\lambda)$ for all $\lambda \geq 0$ and $t \in T$, where $\psi$ is convex and $\psi(0) = \psi'(0) = 0$. Then*

$$\mathbb{P}\left[\sup_{t \in T} X_t \geq \psi^{*-1}(\log |T| + u)\right] \leq e^{-u} \text{ for all } u \geq 0. \tag{66}$$

*In particular, if $X_t$ is $\sigma^2$-subgaussian and $\mathbb{E}[X_t] = 0$ for every $t \in T$, then*

$$\mathbb{P}\left[\sup_{t \in T} X_t \geq \sqrt{2\sigma^2 \log |T|} + x\right] \leq e^{-\frac{x^2}{2\sigma^2}} \text{ for all } x \geq 0. \tag{67}$$

We estimate the tail probability of $X_W$ in the following theorem:

**Theorem 8.** *Let $\{X_t\}_{t\in T}$ be a random process and $T$ a finite set. Assume that $\Lambda_{X_t}(\lambda) \leq \psi(\lambda)$ for all $\lambda \geq 0$ and $t \in T$, where $\psi$ is convex and $\psi(0) = \psi'(0) = 0$, and let $W$ be a random variable taking values on $T$. Then for all $u \geq 0$,*

$$\mathbb{P}\left[X_W \geq \psi^{*-1}(I(W;X_T) + u)\right]$$

$$\leq \min\left\{\frac{I(W;X_T) + \log\left(2 - e^{-I(W;X_T)-u}\right)}{I(W;X_T) + u}, e^{\log|T| - I(W;X_T) - u}\right\}. \tag{68}$$

*In particular, if $X_t$ is $\sigma^2$-subgaussian and $\mathbb{E}[X_t] = 0$ for every $t \in T$, then for all $x \geq 0$,*

$$\mathbb{P}\left[X_W \geq \sqrt{2\sigma^2 I(W;X_T)} + x\right]$$

$$\leq \min\left\{\frac{I(W;X_T) + \log\left(2 - e^{-I(W;X_T)-\frac{x^2}{2\sigma^2}}\right)}{I(W;X_T) + \frac{x^2}{2\sigma^2}}, e^{\log|T| - I(W;X_T) - \frac{x^2}{2\sigma^2}}\right\}. \tag{69}$$

*Proof.* Analogous to the proof of Theorem 7 in [8], [16], we invoke the variational representation of relative entropy (Theorem 6) in our proof.

Define $n \triangleq |T|$ and without loss of generality, let $T \triangleq \{1, 2, ..., n\}$. Note that

$$\mathbb{P}\left[X_W \geq \psi^{*-1}(I(W;X_T) + u)\right]$$

$$= \sum_{i=1}^{n} \mathbb{P}\left[X_W \geq \psi^{*-1}(I(W;X_T) + u)\Big| W = i\right] \mathbb{P}[W = i] \tag{70}$$

$$= \sum_{i=1}^{n} \mathbb{P}\left[X_i \geq \psi^{*-1}(I(W;X_T) + u)\Big| W = i\right] \mathbb{P}[W = i]. \tag{71}$$

Define

$$f(a) \triangleq \zeta \mathbb{1}_{\{a \geq \psi^{*-1}(I(W;X_T)+u)\}}, \tag{72}$$

where $\zeta > 0$ is an arbitrary real number. Choose an arbitrary $1 \leq i \leq n$, and define random variable $X$ such that $P_X = P_{X_i|W=i}$. We have

$$\zeta\mathbb{P}\left[X_i \geq \psi^{*-1}(I(W;X_T) + u)\Big| W = i\right]$$

$$= \mathbb{E}[f(X)] \tag{73}$$

$$\leq D(P_X \| P_{X_i}) + \log \mathbb{E}[e^{f(X_i)}] \tag{74}$$

$$= D(P_{X_i|W=i} \| P_{X_i}) + \log \mathbb{E}[e^{f(X_i)}] \tag{75}$$

$$= D(P_{X_i|W=i} \| P_{X_i}) + \log\left(e^{\zeta}\mathbb{P}\left[X_i \geq \psi^{*-1}(I(W;X_T) + u)\right]\right.$$

$$\left. + \mathbb{P}\left[X_i < \psi^{*-1}(I(W;X_T) + u)\right]\right) \tag{76}$$

$$= D(P_{X_i|W=i} \| P_{X_i})$$

$$+ \log\left((e^{\zeta} - 1)\mathbb{P}\left[X_i \geq \psi^{*-1}(I(W;X_T) + u)\right] + 1\right) \tag{77}$$

$$\leq D(P_{X_i|W=i} \| P_{X_i}) + \log\left((e^{\zeta} - 1)e^{-I(W;X_T)-u} + 1\right) \tag{78}$$

$$\leq D(P_{X_T|W=i} \| P_{X_T}) + \log\left((e^{\zeta} - 1)e^{-I(W;X_T)-u} + 1\right), \tag{79}$$

where (74) is based on Theorem 6, (78) is based on Lemma 4 and (79) is based on the data processing inequality for relative entropy. Therefore

$$\mathbb{P}\left[X_i \geq \psi^{*-1}(I(W;X_T) + u)\Big| W = i\right]$$

$$\leq \frac{1}{\zeta}\left(D(P_{X_T|W=i} \| P_{X_T}) + \log\left((e^{\zeta} - 1)e^{-I(W;X_T)-u} + 1\right)\right). \tag{80}$$

Since $i$ was chosen arbitrarily, (80) holds for all $i = 1, 2, ..., n$. Thus, based on (70) and (71) we have

$$\mathbb{P}\left[X_W \geq \psi^{*-1}(I(W; X_T) + u)\right] \leq \frac{1}{\zeta}\left(\sum_{i=1}^{n} D(P_{X_T|W=i}\|P_{X_T})\mathbb{P}[W = i]\right.$$
$$\left. + \log\left((e^\zeta - 1)e^{-I(W;X_T)-u} + 1\right)\right) \qquad (81)$$
$$= \frac{1}{\zeta}\left(I(W; X_T) + \log\left((e^\zeta - 1)e^{-I(W;X_T)-u} + 1\right)\right). \qquad (82)$$

Since (82) holds for arbitrary $\zeta > 0$, we can infimize the right side of (82) over $\zeta$ to obtain

$$\mathbb{P}\left[X_W \geq \psi^{*-1}(I(W; X_T) + u)\right]$$
$$\leq \inf_{\zeta > 0}\left\{\frac{1}{\zeta}\left(I(W; X_T) + \log\left((e^\zeta - 1)e^{-I(W;X_T)-u} + 1\right)\right)\right\}. \qquad (83)$$

Now, we upper bound the right side of (83) by choosing $\zeta \leftarrow I(W; X_T) + u$, to get

$$\mathbb{P}\left[X_W \geq \psi^{*-1}(I(W; X_T) + u)\right] \leq \frac{I(W; X_T) + \log\left(2 - e^{-I(W;X_T)-u}\right)}{I(W; X_T) + u}, \qquad (84)$$

which is one of the terms in the right side of (68). To prove the other upper bound in (68), note that

$$\mathbb{P}\left[X_W \geq \psi^{*-1}(I(W; X_T) + u)\right] = \sum_{i=1}^{n}\mathbb{P}\left[X_W \geq \psi^{*-1}(I(W; X_T) + u), W = i\right] \qquad (85)$$
$$= \sum_{i=1}^{n}\mathbb{P}\left[X_i \geq \psi^{*-1}(I(W; X_T) + u), W = i\right] \qquad (86)$$
$$\leq \sum_{i=1}^{n}\mathbb{P}\left[X_i \geq \psi^{*-1}(I(W; X_T) + u)\right] \qquad (87)$$
$$\leq ne^{-I(W;X_T)-u} \qquad (88)$$
$$= e^{\log|T|-I(W;X_T)-u}, \qquad (89)$$

where (88) is based on Lemma 4.

For the subgaussian case, note that

$$\psi^{*-1}(\log|T| + u) = \sqrt{2\sigma^2(\log|T| + u)} \qquad (90)$$
$$\leq \sqrt{2\sigma^2\log|T|} + \sqrt{2\sigma^2 u}, \qquad (91)$$

therefore, based on (84) and (89), we get (69). $\qquad\square$

Note that our upper bound in (84) is slightly stronger than Lemma 4.1 in [20], and our method of proving (84) shows that Lemma 4.1 in [20] is a corollary of the well known variational representation of relative entropy (Theorem 6).

**Remark 7.** If the assumptions of Proposition 3 hold, then by applying Theorem 8 on $\{-X_t\}_{t \in T}$, it is straightforward to obtain analogous lower tail bounds for $X_W$.

## C  Lipschitz processes and the $\epsilon$-net argument

The generalization of the maximal inequality (Proposition 1) to random processes with infinite number of random variables is not useful, since its upper bound blows up. But in many applications, there exists some dependence structure between the random variables of the random process which can be exploited to give better bounds. In this section we define *Lipschitz* structure and mention the $\epsilon$-net argument. Then we show how to tighten that by using mutual information.

**Definition 7** (Lipschitz process)**.** *The random process* $\{X_t\}_{t \in T}$ *is called* Lipschitz *for a metric $d$ on $T$ if there exists a random variable $C$ such that $|X_t - X_s| \leq Cd(t, s)$ for all $t, s \in T$.*

Here we give the definitions of $\epsilon$-net and covering number $N(T, d, \epsilon)$:

**Definition 8** ($\epsilon$-net and covering number). *Let $d$ be a metric on the set $T$.*

(a) *A finite set $\mathcal{N}$ is called an $\epsilon$-net for $(T, d)$ if there exists a function $\pi_{\mathcal{N}}$ which maps every point $t \in T$ to $\pi_{\mathcal{N}}(t) \in \mathcal{N}$ such that $d(t, \pi_{\mathcal{N}}(t)) \leq \epsilon$.*

(b) *The covering number for a metric space $(T, d)$ is the smallest cardinality of an $\epsilon$-net for that space, where we denote it by $N(T, d, \epsilon)$. In other words,*

$$N(T, d, \epsilon) \triangleq \inf\{|\mathcal{N}| : \mathcal{N} \text{ is an } \epsilon\text{-net for } (T, d)\}. \tag{92}$$

(c) *An $\epsilon$-net $\mathcal{N}$ for the metric space $(T, d)$ is called minimal if $|\mathcal{N}| = N(T, d, \epsilon)$.*

For Lipschitz processes, the following inequality usually gives better bounds than the maximal inequality (Proposition 1), and it is also referred to as *the $\epsilon$-net argument*:

**Proposition 5** (Lipschitz maximal inequality). *Assume that $\{X_t\}_{t \in T}$ is a Lispschitz process for the metric $d$ on $T$, and $\Lambda_{X_t}(\lambda) \leq \psi(\lambda)$ for all $\lambda \geq 0$ and $t \in T$, where $\psi$ is convex and $\psi(0) = \psi'(0) = 0$. Then*

$$\mathbb{E}\left[\sup_{t \in T} X_t\right] \leq \inf_{\epsilon > 0} \left\{\epsilon \mathbb{E}[C] + \psi^{*-1}\left(\log N(T, d, \epsilon)\right)\right\}. \tag{93}$$

For a proof of Proposition 5 see [9]. The following theorem tightens Proposition 5 by using the mutual information method:

**Theorem 9.** *Assume that $\{X_t\}_{t \in T}$ is a Lipschitz process for the metric $d$ on $T$, and $\Lambda_{X_t}(\lambda) \leq \psi(\lambda)$ for all $\lambda \geq 0$ and $t \in T$, where $\psi$ is convex and $\psi(0) = \psi'(0) = 0$. If for all $\epsilon > 0$, $\mathcal{N}_\epsilon$ is an $\epsilon$-net for $(T, d)$, then*

$$\mathbb{E}[X_W] \leq \inf_{\substack{\epsilon > 0 \\ \mathcal{N}_\epsilon}} \left\{\epsilon \mathbb{E}[C] + \psi^{*-1}(I(\pi_{\mathcal{N}_\epsilon}(W); X_{\mathcal{N}_\epsilon}))\right\}, \tag{94}$$

*where the infimum is over all $\epsilon > 0$ and all $\epsilon$-nets $\mathcal{N}_\epsilon$ of $(T, d)$.*

*Proof.* We have $X_W = (X_W - X_{\pi_{\mathcal{N}_\epsilon}(W)}) + X_{\pi_{\mathcal{N}_\epsilon}(W)}$. Therefore, based on Theorem 7 and Definition 7, we have

$$\mathbb{E}[X_W] = \mathbb{E}[X_W - X_{\pi_{\mathcal{N}_\epsilon}(W)}] + \mathbb{E}[X_{\pi_{\mathcal{N}_\epsilon}(W)}] \tag{95}$$

$$\leq \mathbb{E}[|X_W - X_{\pi_{\mathcal{N}_\epsilon}(W)}|] + \mathbb{E}[X_{\pi_{\mathcal{N}_\epsilon}(W)}] \tag{96}$$

$$\leq \epsilon \mathbb{E}[C] + \psi^{*-1}\left(I(\pi_{\mathcal{N}_\epsilon}(W); X_{\mathcal{N}_\epsilon})\right) \tag{97}$$

$\square$

**Remark 8.** Note that in the infimum in (94), for all $\epsilon > 0$ one can restrict $\mathcal{N}_\epsilon$ to be a minimal $\epsilon$-net to conclude that the right side of (94) is no larger than the right side of (93), due to Lemma 2 and the following inequalities:

$$I(\pi_{\mathcal{N}_\epsilon}(W); X_{\mathcal{N}_\epsilon}) \leq H(\pi_{\mathcal{N}_\epsilon}(W)) \tag{98}$$

$$\leq \log N(T, d, \epsilon). \tag{99}$$

**Proposition 6.** *With the assumptions of Theorem 9, we have*

$$\inf_{\substack{\epsilon > 0 \\ \mathcal{N}_\epsilon}} \left\{\epsilon \mathbb{E}[C] + \psi^{*-1}\left(I(\pi_{\mathcal{N}_\epsilon}(W); X_{\mathcal{N}_\epsilon})\right)\right\} \leq \psi^{*-1}(I(W; X_T)). \tag{100}$$

*Therefore the bound on $\mathbb{E}[X_W]$ given in Theorem 9 is no larger than the bound given in Theorem 7.*

*Proof.* For all $\epsilon > 0$, based on the chain rule of mutual information (or the data processing inequality), we have

$$I(\pi_{\mathcal{N}_\epsilon}(W); X_{\mathcal{N}_\epsilon}) \leq I(\pi_{\mathcal{N}_\epsilon}(W); X_T). \tag{101}$$

Furthermore, the Markov chain $\pi_{\mathcal{N}_\epsilon}(W) \leftrightarrow W \leftrightarrow X_T$ and the data processing inequality for mutual information yield

$$I(\pi_{\mathcal{N}_\epsilon}(W); X_T) \leq I(W; X_T). \tag{102}$$

Lemma 2 along with (101) and (102) conclude

$$\epsilon \mathbb{E}[C] + \psi^{*-1}(I(\pi_{\mathcal{N}_\epsilon}(W); X_{\mathcal{N}_\epsilon})) \leq \epsilon \mathbb{E}[C] + \psi^{*-1}(I(W; X_T)). \tag{103}$$

Letting $\epsilon \to 0$ completes the proof. $\square$

**Remark 9.** If in addition to the assumptions of Theorem 9, we have $\Lambda_{X_t}(-\lambda) \leq \psi(\lambda)$ for all $\lambda \geq 0$ and $t \in T$ (see Corollary 1 for an example), then similar to the proof of Proposition 3, we can prove

$$|\mathbb{E}[X_W]| \leq \epsilon \mathbb{E}[C] + {\psi^*}^{-1}(I(\pi_{\mathcal{N}}(W); X_{\mathcal{N}})). \tag{104}$$

## D   Chaining mutual information

We loosen the "almost sure" Lipschitz condition of the dependencies of the random variables of a process to a "in probability" condition, defined as subgaussian processes:

**Definition 9** (Subgaussian process). *The random process $\{X_t\}_{t \in T}$ on the metric space $(T, d)$ is called* subgaussian *if $\mathbb{E}[X_t] = 0$ for all $t \in T$ and*

$$\mathbb{E}\left[e^{\lambda(X_t - X_s)}\right] \leq e^{\frac{1}{2}\lambda^2 d^2(t,s)} \quad \textit{for all} \quad t, s \in T, \lambda \geq 0. \tag{105}$$

We now state a classical chaining result:

**Theorem 10** (Dudley). *[13]. Assume that $\{X_t\}_{t \in T}$ is a separable subgaussian process on the bounded metric space $(T, d)$. Then*

$$\mathbb{E}\left[\sup_{t \in T} X_t\right] \leq 6 \sum_{k \in \mathbb{Z}} 2^{-k} \sqrt{\log N(T, d, 2^{-k})}. \tag{106}$$

By combining the mutual information method and the chaining method, we obtain the following result:

**Theorem 11.** *Assume that $\{X_t\}_{t \in T}$ is a separable subgaussian process on the bounded metric space $(T, d)$ and let $k_0$ be an integer such that $2^{-k_0} \geq \mathrm{diam}(T)$. Let $\{\mathcal{N}_k\}_{k=k_0+1}^{\infty}$ be a sequence of sets, where for each $k > k_0$, $\mathcal{N}_k$ is a $2^{-k}$-net for $(T, d)$. For an arbitrary $t_0 \in T$, let $\mathcal{N}_{k_0} \triangleq \{t_0\}$. Assume that $W$ is a random variable which takes values on $T$. We have*

*(a)*

$$\mathbb{E}[X_W] \leq 3\sqrt{2} \sum_{k=k_0+1}^{\infty} 2^{-k} \sqrt{I(\pi_{\mathcal{N}_k}(W), \pi_{\mathcal{N}_{k-1}}(W); X_T)}. \tag{107}$$

*(b)*

$$\mathbb{E}\left[|X_W - X_{t_0}|\right] \leq 3\sqrt{2} \sum_{k=k_0+1}^{\infty} 2^{-k} \sqrt{I(\pi_{\mathcal{N}_k}(W), \pi_{\mathcal{N}_{k-1}}(W); X_T) + \log 2}. \tag{108}$$

*Proof.*

(a) Since $2^{-k_0} \geq \mathrm{diam}(T)$, we have $N(T, d, 2^{-k_0}) = 1$, therefore $\mathcal{N}_{k_0}$ is a $2^{-k_0}$-net for $(T, d)$. Note that for any integer $n > k_0$ we can write

$$X_W = X_{t_0} + \sum_{k=k_0+1}^{n} (X_{\pi_{\mathcal{N}_k}(W)} - X_{\pi_{\mathcal{N}_{k-1}}(W)}) + (X_W - X_{\pi_{\mathcal{N}_n}(W)}). \tag{109}$$

Since by the definition of subgaussian processes the process is centered, we have $\mathbb{E}[X_{t_0}] = 0$. Thus

$$\mathbb{E}[X_W] - \mathbb{E}[X_W - X_{\pi_{\mathcal{N}_n}(W)}] = \sum_{k=k_0+1}^{n} \mathbb{E}[X_{\pi_{\mathcal{N}_k}(W)} - X_{\pi_{\mathcal{N}_{k-1}}(W)}]. \tag{110}$$

Note that for every $k > k_0$, $\{X_{\pi_{\mathcal{N}_k}(t)} - X_{\pi_{\mathcal{N}_{k-1}}(t)}\}_{t \in T}$ is a subgaussian process with at most $|\mathcal{N}_k||\mathcal{N}_{k-1}|$ distinct terms, hence a finite process. Based on triangle inequality,

$$d(\pi_{\mathcal{N}_k}(t), \pi_{\mathcal{N}_{k-1}}(t)) \leq d(t, \pi_{\mathcal{N}_k}(t)) + d(t, \pi_{\mathcal{N}_{k-1}}(t))$$
$$\leq 3 \times 2^{-k}. \tag{111}$$

Note that knowing the value of $(\pi_{\mathcal{N}_k}(W), \pi_{\mathcal{N}_{k-1}}(W))$ is enough to determine which one of the random variables of $\{X_{\pi_{\mathcal{N}_k}(t)} - X_{\pi_{\mathcal{N}_{k-1}}(t)}\}_{t \in T}$ is chosen according to $W$. Therefore $(\pi_{\mathcal{N}_k}(W), \pi_{\mathcal{N}_{k-1}}(W))$ is playing the role of the random index, and since $X_{\pi_{\mathcal{N}_k}(t)} - X_{\pi_{\mathcal{N}_{k-1}}(t)}$ is $d^2(\pi_{\mathcal{N}_k}(t), \pi_{\mathcal{N}_{k-1}}(t))$-subgaussian, based on Theorem 7, we have

$$\mathbb{E}\left[X_{\pi_{\mathcal{N}_k}(W)} - X_{\pi_{\mathcal{N}_{k-1}}(W)}\right]$$
$$\leq 3\sqrt{2} \times 2^{-k} \left(I(\pi_{\mathcal{N}_k}(W), \pi_{\mathcal{N}_{k-1}}(W); \{X_{\mathcal{N}_k(t)} - X_{\mathcal{N}_{k-1}(t)}\}_{t \in T})\right)^{\frac{1}{2}}. \tag{112}$$

Based on the chain rule of mutual information, adding random variables to one side of mutual information does not decrease its value. Thus

$$\mathbb{E}[X_{\pi_{\mathcal{N}_k}(W)} - X_{\pi_{\mathcal{N}_{k-1}}(W)}] \leq 3\sqrt{2} \times 2^{-k} \left(I(\pi_{\mathcal{N}_k}(W), \pi_{\mathcal{N}_{k-1}}(W); X_{\mathcal{N}_k} - X_{\mathcal{N}_{k-1}})\right)^{\frac{1}{2}}. \tag{113}$$

From (110) and by using (113) for each $k = k_0 + 1, \ldots, n$, we conclude

$$\mathbb{E}[X_W] - \mathbb{E}[X_W - X_{\pi_{\mathcal{N}_n}(W)}] \leq \sum_{k=k_0+1}^{n} 3\sqrt{2} \times 2^{-k} \left(I(\pi_{\mathcal{N}_k}(W), \pi_{\mathcal{N}_{k-1}}(W); X_{\mathcal{N}_k} - X_{\mathcal{N}_{k-1}})\right)^{\frac{1}{2}}. \tag{114}$$

Note that $|\mathbb{E}[X_W - X_{\pi_{\mathcal{N}_n}(W)}]| \leq \mathbb{E}[\sup_{t \in T}(X_t - X_{\pi_{\mathcal{N}_n}(t)})]$, and since the process is separable, we have

$$\lim_{n \to \infty} \mathbb{E}[\sup_{t \in T}(X_t - X_{\pi_{\mathcal{N}_n}(t)})] = 0, \tag{115}$$

(see proof of Theorem 5.24 in [9].) Hence

$$\lim_{n \to \infty} \mathbb{E}[X_W - X_{\pi_{\mathcal{N}_n}(W)}] = 0. \tag{116}$$

Based on (114) and (116), we get

$$\mathbb{E}[X_W] \leq 3\sqrt{2} \sum_{k=k_0+1}^{\infty} 2^{-k} \left(I(\pi_{\mathcal{N}_k}(W), \pi_{\mathcal{N}_{k-1}}(W); X_{\mathcal{N}_k} - X_{\mathcal{N}_{k-1}})\right)^{\frac{1}{2}}. \tag{117}$$

By further upper bounding the right side of (117), we obtain

$$\mathbb{E}[X_W] \leq 3\sqrt{2} \sum_{k=k_0+1}^{\infty} 2^{-k} \left(I(\pi_{\mathcal{N}_k}(W), \pi_{\mathcal{N}_{k-1}}(W); X_{\mathcal{N}_k} - X_{\mathcal{N}_{k-1}})\right)^{\frac{1}{2}}$$

$$\leq 3\sqrt{2} \sum_{k=k_0+1}^{\infty} 2^{-k} \left(I(\pi_{\mathcal{N}_k}(W), \pi_{\mathcal{N}_{k-1}}(W); X_{\mathcal{N}_k} - X_{\mathcal{N}_{k-1}}, X_{\mathcal{N}_{k-1}})\right)^{\frac{1}{2}} \tag{118}$$

$$= 3\sqrt{2} \sum_{k=k_0+1}^{\infty} 2^{-k} \left(I(\pi_{\mathcal{N}_k}(W), \pi_{\mathcal{N}_{k-1}}(W); X_{\mathcal{N}_k \cup \mathcal{N}_{k-1}})\right)^{\frac{1}{2}} \tag{119}$$

$$\leq 3\sqrt{2} \sum_{k=k_0+1}^{\infty} 2^{-k} \left(I(\pi_{\mathcal{N}_k}(W), \pi_{\mathcal{N}_{k-1}}(W); X_T)\right)^{\frac{1}{2}}, \tag{120}$$

where (118) and (120) follow from the chain rule of mutual information, and (119) follows from the fact that mutual information is invariant to one-to-one functions.

(b) From (109) we conclude that

$$|X_W - X_{t_0}| \leq \sum_{k=k_0+1}^{n} |X_{\pi_{\mathcal{N}_k}(W)} - X_{\pi_{\mathcal{N}_{k-1}}(W)}| + |X_W - X_{\pi_{\mathcal{N}_n}(W)}|. \tag{121}$$

Hence

$$\mathbb{E}[|X_W - X_{t_0}|] - \mathbb{E}[|X_W - X_{\pi_{\mathcal{N}_n}(W)}|] \leq \sum_{k=k_0+1}^{n} \mathbb{E}[|X_{\pi_{\mathcal{N}_k}(W)} - X_{\pi_{\mathcal{N}_{k-1}}(W)}|]. \tag{122}$$

The rest of the proof is similar to previous part, with the difference of instead of using Theorem 7 to obtain (112), we use Proposition 3 (b) with $\psi(\lambda) \triangleq \frac{\lambda^2 \sigma^2}{2}$ to obtain

$$\mathbb{E}\left[|X_{\pi_{\mathcal{N}_k}(W)} - X_{\pi_{\mathcal{N}_{k-1}}(W)}|\right]$$
$$\leq 3\sqrt{2} \times 2^{-k} \left(I(\pi_{\mathcal{N}_k}(W), \pi_{\mathcal{N}_{k-1}}(W); \{X_{\mathcal{N}_k(t)} - X_{\mathcal{N}_{k-1}(t)}\}_{t \in T}) + \log 2\right)^{\frac{1}{2}}. \tag{123}$$

$\square$

**Remark 10.** Note that for all $k > k_0$,

$$I(\pi_{\mathcal{N}_k}(W), \pi_{\mathcal{N}_{k-1}}(W); X_T) \leq H(\pi_{\mathcal{N}_k}(W), \pi_{\mathcal{N}_{k-1}}(W)) \tag{124}$$
$$\leq H(\pi_{\mathcal{N}_k}(W)) + H(\pi_{\mathcal{N}_{k-1}}(W)) \tag{125}$$
$$\leq \log|\mathcal{N}_k| + \log|\mathcal{N}_{k-1}| \tag{126}$$
$$\leq 2\log|\mathcal{N}_k|. \tag{127}$$

Therefore, if we assume that for each $k > k_0$, $\mathcal{N}_k$ is a *minimal* $2^{-k}$-net for $(T,d)$, then we have replaced the Hartley entropy in Dudley's inequality (Theorem 10) with Shannon entropy (because $\log|\mathcal{N}_k| = \log N(T,d,2^{-k})$) and further with mutual information.

We are now able to present the proof of the small subset property theorem:

*Proof of Theoerem 5.* For each $k \geq k_1(T)$, let $\mathcal{N}_k^{(1)}$ and $\mathcal{N}_k^{(2)}$ be minimal $2^{-k}$-nets for $T_1$ and $T_2$, respectively. It is clear that $\mathcal{N}_k \triangleq \mathcal{N}_k^{(1)} \cup \mathcal{N}_k^{(2)}$, is a $2^{-k}$-net for $T$. Let

$$\pi_{\mathcal{N}_k}(t) \triangleq \begin{cases} \pi_{\mathcal{N}_k^{(1)}}(t) & \text{if } t \in T_1 \\ \pi_{\mathcal{N}_k^{(2)}}(t) & \text{if } t \in T_2 \end{cases}.$$

Based on Theorem 11 and Remark 10, we have

$$\mathbb{E}[X_W] \leq 3\sqrt{2} \sum_{k=k_1(T)}^{\infty} 2^{-k} \left( H(\pi_{\mathcal{N}_k}(W)) + H(\pi_{\mathcal{N}_{k-1}}(W)) \right)^{\frac{1}{2}}$$

$$\leq 3\sqrt{2} \sum_{k=k_1(T)}^{\infty} 2^{-k} \left( \alpha \log|\mathcal{N}_k^{(1)}| + (1-\alpha)\log|\mathcal{N}_k^{(2)}| \right.$$

$$\left. + \alpha \log|\mathcal{N}_{k-1}^{(1)}| + (1-\alpha)\log|\mathcal{N}_{k-1}^{(2)}| + 2H(\alpha) \right)^{\frac{1}{2}}$$

$$\leq 3\sqrt{2} \sum_{k=k_1(T)}^{\infty} 2^{-k} \left( \alpha \log|\mathcal{N}_k^{(1)}|^2 + (1-\alpha)\log|\mathcal{N}_k^{(2)}|^2 + 2H(\alpha) \right)^{\frac{1}{2}}$$

$$\leq 6 \sum_{k=k_1(T)}^{\infty} 2^{-k} \left( \alpha \log|\mathcal{N}_k^{(1)}| + (1-\alpha)\log|\mathcal{N}_k^{(2)}| + H(\alpha) \right)^{\frac{1}{2}}$$

$$= 6 \sum_{k=k_1(T)}^{\infty} 2^{-k} \left( \alpha \log N(T_1,d,2^{-k}) + (1-\alpha)\log N(T_2,d,2^{-k}) + H(\alpha) \right)^{\frac{1}{2}}. \tag{128}$$

$\square$

For random processes other than subgaussian processes, where the tail of increments are controlled by a function $\psi$, we have the following result whose proof is similar to the proof of Theorem 11:

**Theorem 12.** *Assume that $\{X_t\}_{t \in T}$ is a separable process defined on the bounded metric space $(T,d)$, with $\mathbb{E}[X_t] = 0$ for all $t \in T$ and*

$$\log \mathbb{E}\left[ e^{\frac{\lambda(X_t - X_s)}{d(t,s)}} \right] \leq \psi(\lambda) \text{ for all } t, s \in T, \lambda \geq 0, \tag{129}$$

*where $\psi$ is convex and $\psi(0) = \psi'(0) = 0$. Let $k_0$ be an integer such that $2^{-k_0} \geq \mathrm{diam}(T)$ and $\{\mathcal{N}_k\}_{k=k_0+1}^{\infty}$ be a sequence of sets, where for each $k > k_0$, $\mathcal{N}_k$ is a $2^{-k}$-net for $(T,d)$. For an arbitrary $t_0 \in T$, let $\mathcal{N}_{k_0} \triangleq \{t_0\}$. Assume that $W$ is a random variable which takes values on $T$. We have*

*(a)*

$$\mathbb{E}[X_W] \leq 3\sqrt{2} \sum_{k=k_0+1}^{\infty} 2^{-k} \psi^{*-1} \left( I(\pi_{\mathcal{N}_k}(W), \pi_{\mathcal{N}_{k-1}}(W); X_T) \right). \tag{130}$$

*(b)*

$$\mathbb{E}\left[|X_W - X_{t_0}|\right] \le 3\sqrt{2} \sum_{k=k_0+1}^{\infty} 2^{-k} {\psi^*}^{-1}\left(I(\pi_{\mathcal{N}_k}(W), \pi_{\mathcal{N}_{k-1}}(W); X_T) + \log 2\right). \quad (131)$$