[Reviews · NeurIPS 2018]

Reviewer 1



Overview: This paper gives new generalization bounds in term of a chaining bound for mutual information. Specifically, Russo and Zou (2015) have shown that one can bound the generalization of any statistical learning algorithm in terms of the mutual information between the distribution over the algorithm's output and the distribution over the sample set. The present work goes further and gives a refined upper bound based on chaining in the sense of the classical Dudley chaining. The idea is to assume that the algorithms's output space admits a sequence of partitions at various scales in some metric (typically based on the data distribution). Under this assumption, the authors upper bound generalization in terms of a Dudley type bound which, at each scale, measures the mutual information between the distribution over the partition element that the algorithm's output lands in and the distribution over the sample set. They provide some small scale (d=2) numerical calculations comparing this bound with others and also show how to bound it in terms of classical covering numbers. Originality and significance: I think the main result from the paper (chaining bounds for mutual information, and application to learning) is fairly novel and should find broader mathematical use. I am happy to accept based on this alone. The main feature missing from this paper is examples / applications. The significance of the proposed analysis techniques would be much easier to evaluate if the authors showed how it can be used to improve or at least recover known rates for standard tasks. EG: What does the method get if I want to use ERM to learn a linear function in d dimensions with the square loss? Table 1 is encouraging but including some simple examples with details worked out would be much more convincing. I do think Theorem 5 is a good step in the direction but it seems like it should not be hard to take the examples a bit further and this would considerably strengthen the result. Currently I think that given the lack of worked out applications, mentioning your results in the context of Zhang et al '17 is a stretch. Along these lines I also want to mention that the discussion of the drawbacks of Rademacher complexity comes across as a bit uninformed, as Rademacher complexity is often used as a component of more sophisticated data- and algorithm-dependent analyses including local Rademacher bounds and margin bounds. Clarity and quality: The paper is well-written and easy to follow. ------------------------- After rebuttal: Decreasing my score to a 6 as my opinion of the novelty has decreased after spending more time with the Audibert-Bousquet paper.

Reviewer 2



The paper begins with musings about deep learning and SGD. It then makes this claim "In particular, classical generalization bounds do not allow to explain this phenomenon." which requires citation as it is not established by this paper, and otherwise it's hearsay. In fact, it's arguably false. On the surface, Bartlett's 1998 bounds say that controlling the norm of the weights suffices and it could have been the case that overparametrized networks allowed for small norm solutions... enough to establish generalization. But, in fact, these bounds imply only that the population classification error is .... less than one. That is, vacuous. Dziugaite and Roy demonstrate this by studying margin bounds based on the path-norm. (Zhang et al, in fact, do not demonstrate that Rademacher bounds cannot explain generalization... no one ever seriously proposed to explain overparametrized neural network generalization by UNIFORM convergence alone!!! Ridiculous.) This papers contributions are more fundamental than recent hype, and so I would suggest cutting the hearsay and sticking to fundamentals as well. I find the conclusion odd as well. I have two concerns: 1. The authors do not seem to be aware that PAC-Bayes bounds relate to mutual information by taking P = E[Q(S)] for S ~ i.i.d. and Q : Z^m \to M(H) the randomized learning algorithm. Then the KL(Q||P) part of the PAC-Bayes bound is equal to the mutual information. While PAC-Bayes bounds control the risk of Gibbs classifiers, taking expectations leads one to bounds on the expected generalization error of a classifier drawn at random, which overlaps with the setting here! This connection with mutual information can be found in Catoni (2007), I believe. It's certainly mentioned by McAllester in one of his PAC-Bayes tutorials too. 2. No citation is made to Audibert and Olivier (2004) "PAC-Bayesian Generic Chaining", which, combined with 1, would seem to have already delivered on the promised combination here, albeit for randomized classifiers. Raginsky et al seem not to be aware of these connections either. I need to hear a concrete plan for how the paper will be modified to address this prior work, which seems to overlap significantly. Other comments: I really liked the example. Extremely clear. Regarding Table 1: the CMI is impressive compared to the chaining and MI bounds on their own, but the bounds seem to be off by 20 or so. What explains that gap? Could a better sequence of partitions have fixed it? Generic chaining? A 20x gap will completely blow any chance of teasing apart the generalization performance of state of the art versus JUNK algorithms. I also like the proof sketch, which is also very very clear. Some issues: 179 The order of operations here is ambiguous. I am parsing this as (argmax_phi X_phi) \oplus Z (mod 2pi) which seems to be the only sensible parsing, but it's annoying I had to spend 20 seconds on this, and manage this uncertainty going forward. 204 strictly speaking, it doesn't fail. it's just vacuous. a failure sounds like the bound is violated, which it isn't. 205 these side facts about LIL and random tournaments are irrelevant. minor typographical issues: 122 hyphens should be an "endash" in Azuma--Hoeffding and other pairings of names. 130 typo: increasing sequence 136 text is bouncing back and forth between "the metric space" and "a (bounded) metric space". Are we now assuming (T,d) is bounded? Or only for this paragraph. If you stick with "the", consider writing "Assume now that (T,d) is a bounded space." 136 unnecessary to recall this soon after making these definitions. 204 the union bound

Reviewer 3



This paper develops chaining methods for the mutual-information-based generalization bounds in Russo and Zou'15. For an increasing sequence of partitions of the metric space, the paper can provide upper bounds on the expectation of loss function at algorithmic output point, using the sum of mutual information terms along a chain of hierarchical partition, as in classical chaining upper bounds in empirical process theory. It is very interesting to see the mutual information bound working in a multi-resolution way under discretization, and being combined with classical chaining techniques. However, this paper is perhaps not strong enough for NIPS: 1. The techniques are standard and simply adapting the classical chaining arguments to this new setting. It follows standard framework in chaining proofs, and applies the Donsker-Varadhan's variational form of KL divergence that has been used in e.g. Xu and Raginsky'17. 2. If the authors are going to convince readers about the usefulness of the discretization method and hierarchical partitioning, they should at least provide some examples where mutual information type bounds are really useful, and see if they get improved. The example provided in this paper, however, is far from satisfactory and actually very artificial. It's basically adding some atom to mess up the original mutual information bound without discretization, and showing that the chaining still works. It provides very little intuition about what we can get by chaining mutual information. Detailed comments: 1. The authors should have noted existing works on PAC-Bayes chaining results (Audibert and Bousquet, JMLR 2007, Combining PAC-Bayesian and Generic Chaining Bounds). Though the mutual information bounds are different from PAC-Bayes, they are derived from similar techniques. 2. The statement about connection to Zhang et al., (2017) and generalization error of SGD is vacuous and deviates from the theme of this paper. I suggest the authors to remove this part.